# Altered Brain Expression of DNA Methylation and Hydroxymethylation Epigenetic Enzymes in a Rat Model of Neuropathic Pain

**DOI:** 10.3390/ijms24087305

**Published:** 2023-04-15

**Authors:** Diogo Rodrigues, Clara Monteiro, Helder Cardoso-Cruz, Vasco Galhardo

**Affiliations:** 1Departamento de Biomedicina–Unidade de Biologia Experimental, Faculdade de Medicina, Universidade do Porto, 4200-319 Porto, Portugal; diogorafaelrodrigues@hotmail.com (D.R.); cmonteir@med.up.pt (C.M.); hcruz@med.up.pt (H.C.-C.); 2i3S/IBMC, Instituto de Investigação e Inovação em Saúde e Instituto de Biologia Molecular e Celular, Pain Neurobiology Group, Universidade do Porto, 4200-135 Porto, Portugal

**Keywords:** pain, brain, rat model, neuropathic pain, SNI, gene expression, epigenetics

## Abstract

The role of epigenetics in chronic pain at the supraspinal level is yet to be fully characterized. DNA histone methylation is crucially regulated by de novo methyltransferases (DNMT1-3) and ten-eleven translocation dioxygenases (TET1-3). Evidence has shown that methylation markers are altered in different CNS regions related to nociception, namely the dorsal root ganglia, the spinal cord, and different brain areas. Decreased global methylation was found in the DRG, the prefrontal cortex, and the amygdala, which was associated with decreased DNMT1/3a expression. In contrast, increased methylation levels and mRNA levels of TET1 and TET3 were linked to augmented pain hypersensitivity and allodynia in inflammatory and neuropathic pain models. Since epigenetic mechanisms may be responsible for the regulation and coordination of various transcriptional modifications described in chronic pain states, with this study, we aimed to evaluate the functional role of TET1-3 and DNMT1/3a genes in neuropathic pain in several brain areas. In a spared nerve injury rat model of neuropathic pain, 21 days after surgery, we found increased TET1 expression in the medial prefrontal cortex and decreased expression in the caudate-putamen and the amygdala; TET2 was upregulated in the medial thalamus; TET3 mRNA levels were reduced in the medial prefrontal cortex and the caudate-putamen; and DNMT1 was downregulated in the caudate-putamen and the medial thalamus. No statistically significant changes in expression were observed with DNMT3a. Our results suggest a complex functional role for these genes in different brain areas in the context of neuropathic pain. The notion of DNA methylation and hydroxymethylation being cell-type specific and not tissue specific, as well as the possibility of chronologically differential gene expression after the establishment of neuropathic or inflammatory pain models, ought to be addressed in future studies.

## 1. Introduction

Chronic pain is recognized as a worrying public health problem with severe psychological and physical consequences. In the U.S. and Europe, an estimated 20.4% and 19% of their adult populations experience chronic pain, respectively [1,2]. It is associated with a multitude of comorbidities, such as depression, anxiety, cognitive impairment, memory defects, and loss of motivation. Now considered a multidimensional disorder with the involvement of sensorimotor, emotional, and cognitive aspects [3], some of the brain areas with resulting MRI modifications from chronic pain conditions in humans include the hippocampus, the amygdala, the perirhinal, and the prefrontal cortex [4].

Several brain pathways, regions, and networks have been suggested as being relevant in the development and maintenance of chronic pain. Of note, the prefrontal cortex and areas within the limbic system such as the anterior cingulate cortex, the amygdala, the ventral tegmental area, and the nucleus accumbens are associated with the affective aspects of pain and regulate emotional and motivational responses [5]. The progression from acute to chronic pain at the brain level is now understood as a transition from a nociceptive state to an emotional one [6] through a shift in brain activity from areas such as the insula, the anterior cingulate cortex, the thalamus, and the basal ganglia to areas such as the mPFC, the amygdala, and the hippocampus [7].

Chronic pain was suggested to develop due to the persistence of pain memory and the inability to extinguish the memory of pain following injury. An aberrant mPFC–amygdala interaction is projected as key to the understanding of the emotional aspect of chronic pain [7]. In chronic pain, impaired mPFC function is well documented and is associated with cognitive deficits [8], while the amygdala suffers from dysfunctional plasticity and hyperactivity in pain states [7], and its connection with the mPFC provides emotional backing to executive functions [9]. For example, the relation between pain and cognitive impairment was evidenced to be caused by prefrontal cortical deactivation through the amygdala [3].

The hippocampus showed learning and memory deficits and altered plasticity and neurogenesis in neuropathic patients and animals [10]. Cognitive-associated deficits in learning and memory are related to the dorsal hippocampus; emotion- and mood-associated functions, to the ventral hippocampus [11]. It was suggested that plasticity within the hippocampus might be partly responsible for memory and the negative affective aspects of pain [7,12].

The lateral hypothalamus has also been implicated in inflammatory and neuropathic pain, producing analgesia and anti-nociception effects [13,14], while also enacting an essential part in the brain reward system [14].

Rats subjected to a model of neuropathic pain had a decrease in gray matter associated with the development of hypersensitivity to sensory stimuli (coincident with the onset of anxiety-like behaviors) and the level of mechanical hyperalgesia (correlated with reduced volume in areas involved in the sensory and affective dimensions of pain, such as the somatosensory cortex, the anterior cingulate cortex, and the PFC [4]). In the PFC, this model of neuropathic pain was followed by transcriptome-wide changes in gene expression, possibly altering several functional gene networks [15].

For many years, several studies have shown the importance of epigenetic mechanisms in the mediation of pain-related inflammatory conditions at the dorsal root and spinal levels, and those continue to be the best-described roles of epigenetic mechanisms in the development of pain conditions [16]. In contrast, much less is currently known about the supraspinal role of epigenetics in the induction and maintenance of chronic pain. As it happens in other chronic pathological processes, it is probable that epigenetic alterations are critical to the development of pain-related changes at the cellular and structural level observed in brain areas. Since it is currently thought that the temporal dynamics of brain synaptic malplasticity are a major component in the development of chronic pain [17], the crucial role that epigenetic processes play in the facilitation or inhibition of specific brain genes highlights the importance of our understanding of supraspinal epigenetics [18]. In fact, the participation of epigenetics in several pain conditions has been a growing topic in recent years, as shown by the various recent reviews about this topic at both the spinal and supraspinal levels [19,20,21,22].

DNA methylation, an epigenetic mechanism through which the regulation of long-term stable silencing of genes occurs, has a vital role in the transcriptional regulation of physiological and pathological processes [23]. It happens at position 5 of the pyrimidine ring of the cytosine nucleotide (5 mC). In mammals, these reactions are catalyzed by de novo DNA methyltransferases 3a and 3b (DNMT3a/3b) and are sustained by DNMT1 [24]. It has been demonstrated that 5 mC is distributed almost exclusively on palindromic dinucleotides in a symmetrical fashion (CpG sites) [25]. One of the exceptions where methylation is not restricted to CpG are mature neurons [26].

Evidence has shown that 5 mC is implicated in several nociceptive pathways in various locations, namely the dorsal root ganglia, the spinal cord, and distinct pain-associated brain areas [27]. Indeed, in a complex Freund’s adjuvant-induced inflammation model, a reduction in the level of global DNA methylation was found in rat trigeminal ganglia, accompanied by a decrease in DNMT1 and DNMT3a expressions and by findings of several pro-nociceptive genes regulated by DNA methylation, with DNMT3a showing a critical role [28]. Evidence showed an association between decreased global methylation in the PFC and the amygdala (but not in the visual cortex or the thalamus) and between decreased mechanical and thermal sensory thresholds and increased anxiety in neuropathic mice [29].

The complementary mechanism of methylation is demethylation, in which reactions are catalyzed by ten-eleven translocation methylcytosine dioxygenase 1 (TET1), TET2, and TET3, capable of actively oxidizing 5-methylcytosine (5 mC) to 5-hydroxymethylcytosine (5 hmC), a critical intermediary step in the demethylation pathways [30]. DNA and histone methylation were proposed to be coordinated by writers (DNMT1-3b), erasers, and readers [31]. TET2 and especially TET3 are significantly transcribed in the human brain, namely in the cerebellum, the cortex, and the hippocampus [32].

Central nervous system tissues are notably enriched in 5 hmC, showing ten times more relative abundance than in ES cells [33]. Single-base resolution studies have shown that 5 hmC is depleted in CpG islands and enriched in non-CG sites, being increased in gene bodies and depleted in intergenic regions, proximal promoters, and Transcription Start Sites (TSS) [34,35]. Despite its essential role in oxidizing 5 mC to 5 hmC, TET may have a more complex role in transcription regulation, depending on its location inside the gene, and may alter transcription through the recognition of DNA-binding proteins [25,36,37]

Data indicate that 5 hmC is relatively stable in the genome [35] and thus can potentially influence gene transcription through its interaction with reader molecules such as MeCP2 [37]. Many neuropsychiatric pathologies were found to be associated with 5 hmC changes, such as Rett syndrome, Alzheimer’s disease, Huntington’s disease, FXTAS, ataxia–telangiectasia, schizophrenia, bipolar disorder, depression, autism, and intracerebral hemorrhage [32].

In this study, we aimed at testing if the altered patterns in methylation and demethylation gene expression shown for chronic pain at the spinal level are reproducible at the brain level and if these correlate with changes already inferred for other conditions and in knockout models.

## 2. Results

We assessed whether the establishment of neuropathic pain would affect the mRNA levels of key genes related to epigenetic regulation in several brain areas. For this, two groups of adult male rats were anesthetized and subjected to either the spared nerve injury (SNI) model [38] of sciatic nerve neuropathic pain (SNI group, *n* = 7) or to a sham intervention with a similar degree of skin incision and exposure of the sciatic nerve but without any nerve injury (sham group, *n* = 7). The onset and development of neuropathic pain were quantified in each animal at days 7, 14, and 21 after surgery using the von Frey method of assessment of mechanical allodynia [39]. Twenty-one days after the initial surgery, the animals were deeply anesthetized and quickly decapitated for the collection of tissue samples from six brain areas: the dorsal hippocampus, the medial prefrontal cortex, the caudate-putamen, the amygdala, the medial thalamus, and the lateral hypothalamus (Figure 1a).

When compared with the control animals, all SNI-treated rats included in this study developed mechanical allodynia, as indicated by a significant decrease in the mechanical force needed to evoke paw withdrawal induced by von Frey filament stimulation (KW = 33.27, *p* < 0.0001; sham vs. SNI: days 7, 14, 21, *p* < 0.01; Dunn’s post hoc test; Figure 1b).

The expression of TET1 in the adult male rats subjected to the spared nerve injury model of neuropathic pain was, interestingly, shown to be decreased in the caudate-putamen (sham: 0.004244 ± 0.0009151; SNI = 0.001346 ± 0.0001520; t(11) = 2.885, *p* = 0.0148) and the amygdala (sham: 0.003853 ± 0.0006716; SNI = 0.002078 ± 0.0003602; t(11) = 2428, *p* = 0.0335), and increased in the medial prefrontal cortex (sham: 0.005314 ± 0.0006008; SNI = 0.009555 ± 0.0009866; t(10) = 3672, *p* = 0.0043), with no statistically significant changes in the dorsal hippocampus (sham: 0.008510 ± 0.0007772; SNI = 0.007806 ± 0.0003091; t(11) = 0.8919, *p* = 0.3915) or the lateral hypothalamus. The expression of TET2 was only altered in comparison to the rats subjected to sham surgery in the rat medial thalamus, showing a statistically significant increase (sham: 0.007972 ± 0.0007251; SNI = 0.01219 ± 0.001728; t(11) = 2378, *p* = 0.0366) (Figure 2).

From all the studied areas, TET3 was only evidenced to be increased in the medial prefrontal cortex (sham: 0.02374 ± 0.003011; SNI = 0.01604 ± 0.0009623; t(10) = 2439, *p* = 0.0349) and the caudate-putamen (sham: 0.02262 ± 0.003046; SNI = 0.01280 ± 0.0008655; t(12) = 3101, *p* = 0.0092) (Figure 2).

We also observed a decrease in DNMT1 mRNA levels in the medial thalamus (sham: 0.01123 ± 0.001762; SNI = 0.006371 ± 0.0006981; t(12) = 2562, *p* = 0.0249) and the caudate-putamen (sham: 0.01746 ± 0.003977; SNI = 0.007195 ± 0.0003122; t(12) = 2572, *p* = 0.0244). Surprisingly, DNMT3a did not show any statistically significant change in any of the studied areas, adding to the current debate of whether these are relevant to the genesis and maintenance of neuropathic pain [40] (Figure 2).

## 3. Discussion

Several previous studies have shown evidence for the role of DNMT1 and DNMT3a methylation in neuropathic pain. In a chronic constriction injury mouse model, significant DNMT3a upregulation was demonstrated in the spinal cord, with its specific inhibition associated with decreased thermal hyperalgesia [41]. DNMT1, with an established role in canonical methylation activity but also with recognized de novo methylation actions, and DNMT3a, through repression of Kcna2, were found upregulated in the DRG after peripheral nerve injury in mice, and their inhibition or knockout diminished pain hypersensitivity [42,43]. However, a recent study challenged these conclusions, with the data not bestowing clear evidence of protein expression and activity of DNMT3a or DNMT3b in adult primary sensory neurons in a CFA model of inflammatory pain [40].

In contrast, Bai et al. showed a reduction in the expression of DNMT1 and DNMT3a in trigeminal ganglia under inflammatory conditions [28], and in consonance with this, we found a diminished expression of DNMT1 21 days after SNI injury in the medial thalamus and the caudate-putamen and, overall, no altered mRNA expression of DNMT3a in the remaining studied brain areas. These results reinforce the need for a better characterization of these epigenetic markers in a cell-type- and time-dependent manner.

The notion of DNA methylation being cell-type specific and not tissue specific [40] and the possibility of differential time-dependent expression ought to be better addressed in future studies.

Since the discovery of ten-eleven methylcytosine dioxygenases, a new focus on understanding these enzymes’ role in nociception has emerged. Pain studies have mainly focused on its importance at the spinal cord level, with increasing expression levels of TET1 and TET3 associated with increased pain hypersensitivity and allodynia in inflammatory and neuropathic pain models [27,44]. Some causal mechanisms for these findings were proposed and include the hydroxymethylation of promoters or micro-RNAs associated with critical genes and proteins involved in pain (such as KCNA2, STAT3, and BDNF) and the inhibition of DNMTs in binding CpG in DNA. TET1 is also increasingly perceived as an essential player in neural plasticity at the brain level, upregulating genes involved in memory and altering contextual fear memory in the hippocampus [45]. TET3, the most abundant of the three TET dioxygenases in the brain [46], has been shown to be relevant in modulating anxiety behavior in the hippocampus and the prefrontal cortex [47].

In fact, six of the eight statistically different changes in the expression of the epigenetic genes reported in this study occur in the critical prefrontal–limbic–amygdala region (although it should also be noted that several putatively important areas such as the cingulate, the insula, and the ventral hippocampus, among others, were not evaluated here but will certainly be measured in future studies). The importance of the prefrontal–limbic–amygdala pathways in the genesis and/or maintenance of chronic pain states has been the topic of several seminal reviews [6,7,9,48,49].

As hypothesized elsewhere, epigenetic mechanisms may be responsible for regulating and coordinating the various transcriptional modifications described in chronic pain states [50]. The transition to 5 hmC catalyzed by TET showed a critical role in various pathological conditions. It was indeed suggested that abnormal levels of 5 hmC in the blood and the spinal cord are vital markers of CNS dysfunction in the context of nociception. TET1 and TET3 (but not TET2) displayed increased expression in the spinal cord in a model of acute inflammatory pain, and TET3 showed upregulation in dorsal root ganglia neurons and glia after SNI injury. In both models, increasing levels of 5 hmC were found after injury, thereby reinforcing the relationship between TET3 expression and the regulation of nociception [44,51]. It was also demonstrated that TET1/3 knockdown in spinal neurons decreased pain behavior, showing the explicit role of these enzymes in inflammatory nociception [27]. The role of TET enzymes was also studied in supraspinal structures in areas involved in emotion and cognition. Following TET3 ablation, heightened anxiety, decreased spatial orientation, and overactivation of the HPA axis were found, as well as a high number of genes with altered transcription, particularly in the ventral hippocampus, thus evidencing the importance of TET3 in pain-related aspects such as anxiety and cognitive function in mice [47]. Very recent studies have established a role for TET2 in normal and pathological brain function. An induced overexpression of TET2 in the hippocampus was associated with heightened cognitive function and rescued age-related cognitive decline [52]; increased TET2 expression was found to regulate microglial pro-inflammatory responses [53]; TET2 was linked to the regulation of genes associated with depression-like behavior in mice [54].

Nevertheless, contrasting with what was observed in the spinal cord of animals with neuropathic or inflammatory pain, 21 days after the SNI procedure, we found TET1 to be downregulated in the amygdala and the caudate-putamen and upregulated in the medial prefrontal cortex, the latter being congruent with the pattern of reduced global methylation already observed here [29]. Despite the absence of statistically significant changes in DNMT expression in the amygdala and the mPFC, the opposite variation in TET1 mRNA levels in these two brain areas emphasizes the need to further study the involvement of epigenetics in the functional modulating effects of the amygdala on mPFC function in chronic pain [3]. We also showed TET2 to be upregulated in the medial thalamus, as well as decreased mRNA expression of TET3 in the medial thalamus and the caudate-putamen. This suggests a more complex functional role and regulation of these genes in different brain areas in the context of neuropathic pain. Overall, these results can be explained by a dynamic variation in the expression of these genes in distinct brain regions over time in an orderly fashion.

In order to better discern the changes in expression of these genes, it is compelling to analyze their expression in earlier phases after the establishment of neuropathic pain following SNI, such as 5, 10, and 21 days after SNI. In our study, the assessment at day 21 after SNI may already demonstrate markers of long-term genetic alterations, with variable biological implications. A comparison with other models of chronic pain is also warranted to assess if some of these changes also occur in non-neuropathic, inflammatory pain models such as the intraplantar complex Freund’s adjuvant (CFA).

It has been suggested elsewhere that strategies targeting DNA methylation, such as using the demethylation agent 5′-aza-2′-deoxycytidine, may be useful in attenuating nociception by decreasing global methylation and decreasing neural sensitization in the spinal cord [55]. The complex, cell-type-specific dynamics of methylation and demethylation in the brain and the spinal cord ought to be undertaken to deepen our understanding of these broad-acting agents’ therapeutic potential.

## 4. Materials and Methods

Experiments were performed in adult Sprague-Dawley male rats (weight 275–325 g; Charles River Laboratories, Saint-Germain-Nuelles, France). Before surgery, rats were housed in collective standard cages containing environmental enrichment (2 animals per box) and kept on a 12 h light/dark cycle with constant-controlled temperature (21 +/− 1 °C) and humidity (50 +/− 5%) and ad libitum feeding and hydration regimen. All experimental procedures were performed at approximately the same time each day during the cycle’s light period. Rats were habituated to being handled by the experimenters before the start of any experimental procedures.

### 4.1. Spared Nerve Injury

Each animal was subjected to the spared nerve injury (SNI) model of neuropathic pain [38] (referred to as the SNI group, *n* = 7) or to a sham intervention (the sham group, *n* = 7). SNI surgery consists of ligation and transaction of the tibial and common peroneal branches of the sciatic nerve while sparing the sural nerve.

Rats were anesthetized with a ketamine/medetomidine mixture (75 and 0.5 mg/kg in saline, respectively, i.p.) and subjected to the SNI model or a sham intervention involving the same extent of skin and muscle dissection, with the exposition of their sciatic nerves, but with no further nerve manipulation.

Evaluation of the sensory threshold for noxious mechanical stimulation was measured at 7, 14, and 21 days after the surgery using von Frey filaments (Somedic, Sösdala, Sweden), as previously described [39,56]. Briefly, von Frey testing was always performed during the light phase in an elevated chamber with a thin metallic mesh floor that allowed easy access to the plantar surface of the hind paw. Filament series were run from the thinnest to the widest to detect the filament from which the animal withdrew the paw in at least 6 out of 10 successive applications.

### 4.2. Sample Collection

For gene expression analysis, rats were deeply anesthetized by intraperitoneal injection of 200 mg/kg sodium pentobarbital solution on day 21 after performing SNI. Brains were immediately removed after decapitation, bathed in RNALater (Ambion) for RNA preservation, and kept refrigerated during further processing. Thick coronal slabs were cut, and individual brain areas were then carefully dissected under a surgical microscope, resulting in bilateral brain samples from several brain areas from 14 animals.

All tissue samples were flash-frozen in liquid nitrogen and stored at −80 °C until further processing. The samples further analyzed corresponded to the contralateral side of the SNI lesion or sham.

### 4.3. RNA Extraction and RT-PCR

After mechanical tissue disruption, RNA extraction was performed using TRIzol^®^ reagent with PureLink^®^ RNA Mini Kit (Life Technologies, Carlsbad, CA, USA). RNA was treated with DNase I (Qiagen, Hilden, Germany) to avoid DNA contamination. Quality and yield of total RNA were assessed through NanoDrop™ analysis. A total of 1 µg of total RNA extracted from frozen tissue was reverse-transcribed in a volume of 20 µL using qScript^®^ cDNA SuperMix kit (QuantaBio, Beverly, MA, USA) according to the manufacturer’s instructions.

RT-qPCR was performed using StepOnePlusTM Real-Time PCR System (Life-Technologies) with PowerUp SYBR Green Master Mix (Thermo Fisher Scientific, Waltham, MA, USA), gene primers (0.4 µM) and cDNA. All genes of interest were normalized to the glyceraldehyde 3-phosphate dehydrogenase (GAPDH) reference gene. Real-time PCRs were performed in triplicate. Primer sequences are listed in Table 1.

Reactions consisted of a 4 min holding stage (2 min at 50 °C and then 2 min at 95 °C), followed by 40 cycles at 95 °C for 30 s, 60 °C for 45 s, and 72 °C for 45 s. The primers were designed using the Primer-BLAST tool to be exon-spanning, and they were initially validated using a SYBR green RT-qPCR assay, which allowed for melting temperature analysis, demonstrating a single peak for each gene product.

### 4.4. Data Analysis

For each animal, the average Ct values for each sample were calculated, and samples were grouped by SNI against control in each brain area. For each sample, the semi-quantitative expression of the gene of interest was performed according to the delta Ct method, using GAPDH as housekeeping gene. All sample groups were tested for normal distribution using both the Shapiro–Wilk test and the Kolmogorov–Smirnov test of sample normality. Only 5 of the 60 group samples had non-normal distribution, requiring their group-wise comparisons to use the Mann–Whitney non-parametric unpaired test instead of the t-test of unpaired parametric comparisons. All the significant differences in gene expression between experimental groups reported here were obtained from normal groups, and for that reason, all were assessed using t-test (GraphPad Prism 8.0 software). All averaged values are given as the mean ± SEM. Statistical significance was accepted at a level of 0.05.

## Figures and Tables

**Figure 1 ijms-24-07305-f001:**
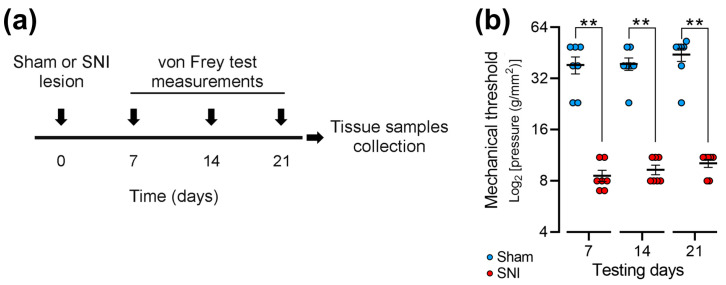
Experimental timeline and mechanical sensitivity threshold. (**a**) Experimental protocol timeline. Briefly, each rat was subjected to a unilateral sham or spared nerve injury (SNI) lesion. Brain tissue samples were collected 21 days after lesion. (**b**) Level of mechanical sensitivity measured by withdrawal response to von Frey filament stimulation across different time points. As expected, a large decrease was observed in the threshold required to induce a paw response in the SNI-treated rats (*n* = 7) when compared with control (sham)-treated rats (*n* = 7). Comparisons between experimental groups and treatments are based on the non-parametric Kruskal–Wallis test followed by Dunn’s post hoc test. Values are presented as mean ± S.E.M. **, *p* < 0.01.

**Figure 2 ijms-24-07305-f002:**
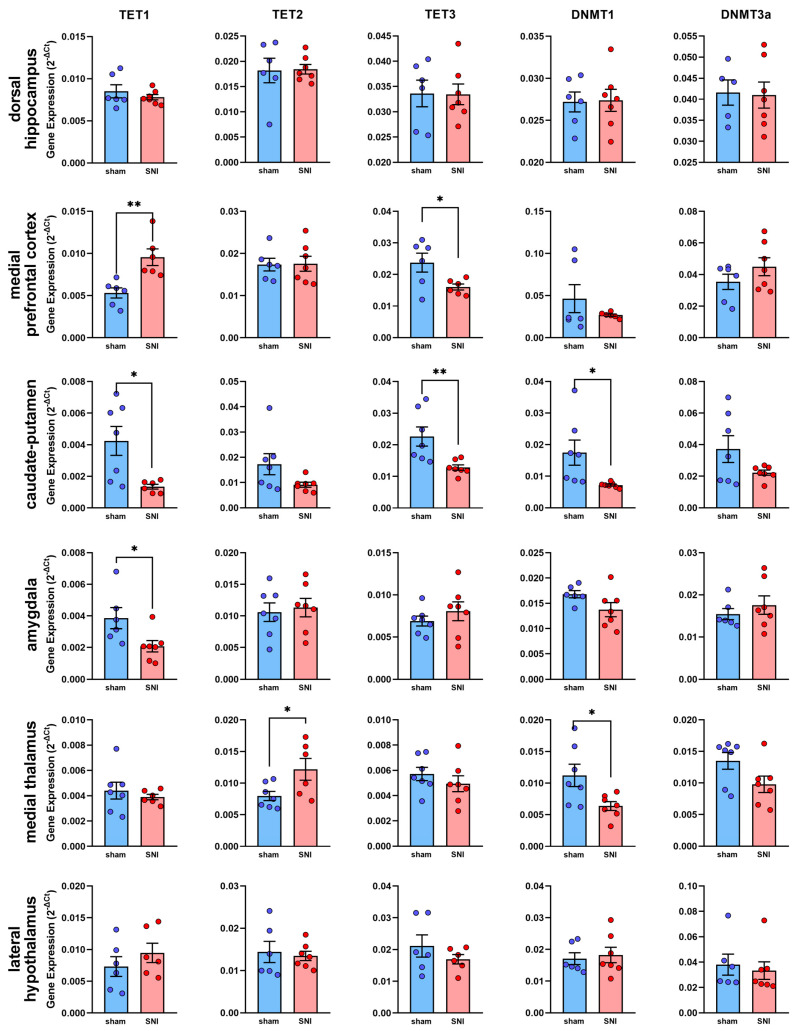
Gene expression in the dorsal hippocampus, medial prefrontal cortex, caudate-putamen, amygdala, medial thalamus, and lateral hypothalamus, contralateral to the SNI neuropathic pain model lesion (*n* = 7 per group). Gene expression was assessed by real-time PCR using GAPDH as housekeeping gene. Each bar represents the average of individual tissue samples run separately from both control (sham) and pain animals (SNI). Statistical analyses were performed using the t-Student parametric test or the Mann–Whitney non-parametric test, depending on the normality of the group sample. TET1-3, ten-eleven translocation methylcytosine dioxygenase 1-3; DNMT1/3a, de novo DNA methyltransferases 1/3a. *, *p* < 0.05; **, *p* < 0.01.

**Table 1 ijms-24-07305-t001:** List of primer sequences used in the study.

Gene	Accession Number	Forward Primer Sequence	Reverse Primer Sequence	Product Length
TET1	ENSRNOG00000000277.5	5′-ACAATGGAAG CACTGTGGTTT-3′	5′-CAGTGTCTGC AAGCCGGTAT-3′	111 bp
TET2	ENSRNOG00000023579.4	5′-GTCGAGTTTG AACACCGAGC-3′	5′-GTGACCACCA CTGTACTGCC-3′	146 bp
TET3	ENSRNOT00000031312.3	5′-AGAACCAGGTG ACCAATGAGG-3′	5′-CAGTGCACCCA TTGTAGAGGT-3′	140 bp
DNMT1	NM_053354.3	5′-GGAGCAAGTC GGACAGTGAG-3′	5′-CGTTTAGCGG GACCCTTGAA-3′	112 bp
DNMT3a	NM_001003958.1 ex 10	5′-TCGCCAATAAC CACGACCAGGA-3′	5′-AGGAGCCCTG TAGCAATCCCA-3′	118 bp
GAPDH	NM_017008.4	5′– GCCATCAACG ACCCCTTCAT-3′	5′-TTCACACCCA TCACAAACAT-3′	314 bp

## Data Availability

The data presented in this study are available upon request from the corresponding author.

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
