# Peer review of "Altered Brain Expression of DNA Methylation and Hydroxymethylation Epigenetic Enzymes in a Rat Model of Neuropathic Pain"

_ijms, 2023, doi:10.3390/ijms24087305_

Round 1

Reviewer 1 Report

Diogo Rodrigues et al performed a study to investigate changes in DNA methylation and hydroxymethylation epigenetic enzyme expression in the brains of mice with neuropathic pain (SNI model). However, there are some that require clarification and improvement in the manuscript.

(1)     The concept of epigenetics, including DNA methylation, has been extensively discussed in pain studies. Therefore, the authors need to strengthen the background of their study by providing a more comprehensive literature review.

(2)    The authors could perform a correlation analysis to determine the relationship between pain-like behaviors in mice and tissue expressions. This could help to establish the strength of the correlation between the analyzed protein expressions and the observed behaviors.

(3)    The expression of TET1 was found to differ across different brain areas. The authors should provide a detailed explanation of this phenomenon in the discussion section of the manuscript. Furthermore, the authors may also compare their findings with other studies that have investigated models of pain to further contextualize the results.

Author Response

The authors thank the reviewer for the relevant comments made about our manuscript. Regarding the reviewer comments:

(1) As suggested, we have now included in the Introduction additional information and references about epigenetics in pain studies.

(2) A correlation analysis to determine the relationship between pain-like behaviors in mice and tissue expressions is certainly a future follow-up study. However in this dataset, the only assessed behaviour was the mechanical threshold of sensitivity to Von Frey filaments, and those results are not adequate for a correlational analysis because all the animals have basically the same mechanical threshold thus negating the possibility of a correlational analysis. As can be observed in Figure 1b, all SNI animals end up being sensitive to only 3 consecutive nylon filaments of the Von Frey test set. However, throughout the 3 tests (at 7, 14 and 21 days), their response varied, and only two animals were consistent across the VonFrey tests: the majority were sensitive to the first or second filement ( 7 or 8 grams, respectively) and then tested sensitive to the third filement on day 21. Furthermore, the protocol of the Von Frey test makes the interpretation even less straigthforward because a positive response is noted to the filament to which the animal withdraws the paw in at least 6 of 10 applications. Thus, an animal is marked as responding to 11 grams when it responds 7 times to the 11g filament, despite the fact that it may respond 5 times to the 8g and 4 times to the 7g filament. Therefore, for these reasons and to the best of our knowledge the Von Frey test is never used in correlations at the INDIVIDUAL level (only for group comparisons).

We have briefly expanded this section in the Materials and Methods section, in order to better explain the Von Frey protocol.

The correlation analysis that the reviewer suggests requires a different behavioral test with better individual resolution and consistency, with gene expression being evaluated at different at each separate time-point. This experiment is already planned for future studies.

(3) To the best of our knowledge, no previous study has quantified the level of TET1-3 gene expression across brain regions in preclinical or clinical models of chronic pain. Thus, there is little that we can add to the discussion without being highly speculative. As suggested, we have briefly expanded our discussion to integrate the roles of crucial brain regions in pain chronification.

Reviewer 2 Report

Dear. Authors, 

This manuscript is well-structured, and its logical development has no critical defect. In addition, the conclusions drawn from this study provide helpful and analytical information about neuropathic pain.

However, when the abbreviation was first described, the authors made no explanation. (Example: CFA in line 90, SNI in line 126) The abbreviation is explained later in the manuscript. I recommend correcting these minor mistakes.

Thank you.

Author Response

The authors thank the reviewer for the praising comments made about our manuscript. As suggested, we have corrected the first-appearance of abbreviations.

Reviewer 3 Report

The authors treaat a very interesting topic about altered brain expression of DNA methylation and hydroxymethylation epigenetic enzymes in a rat model of neuropathic pain. The role of endogenic factors in several pathologies is well known (https://pubmed.ncbi.nlm.nih.gov/27874938/).

I just ask if matherials and methods could be put as second paragaraph as usual. 

Author Response

The authors thank the reviewer for the praising comments made about our manuscript.

The reviewer suggests a reference on epigenetics on several pathologies (https://pubmed.ncbi.nlm.nih.gov/27874938/). However, there must be a mistake in the copy-paste of that particular reference, because the pubmed link refers to a paper on the RANKL pathway (not an epigenetic factor, although a target like all other genes) in cardiac ischemia.

The Material and Methods section appears in the end of the manuscript because that is the normal order of appearance of sections in the IJMS journal, and it is not a decision of the authors.

Round 2

Reviewer 1 Report

The authors have addressed the comments from reviewers and taking time to revise manuscript accordingly. The reviewer have carefully checked and pleased to accept the current form of the manuscript.